# Targeting Apoptosis in AML: Where Do We Stand?

**DOI:** 10.3390/cancers14204995

**Published:** 2022-10-12

**Authors:** Kinga Krawiec, Piotr Strzałka, Magdalena Czemerska, Aneta Wiśnik, Izabela Zawlik, Agnieszka Wierzbowska, Agnieszka Pluta

**Affiliations:** 1Department of Hematology, Medical University of Lodz, 93-513 Lodz, Poland; 2Copernicus Multi-Specialist Oncology and Traumatology Center, 93-513 Lodz, Poland; 3Institute of Medical Sciences, College of Medical Sciences, University of Rzeszow, 35-310 Rzeszow, Poland; 4Laboratory of Molecular Biology, Centre for Innovative Research in Medical and Natural Sciences, College of Medical Sciences, University of Rzeszow, 35-310 Rzeszow, Poland

**Keywords:** acute myeloid leukemia, apoptosis, neddylation, BCL-2 family

## Abstract

**Simple Summary:**

In patients with acute myeloid leukemia (AML), genetic mutations can cause cells to evade regulated cell death (RCD), resulting in excessive cell proliferation. The best-known form of RCD is apoptosis, which prevents the emergence of cancer cells; disturbances in this process are an important factor in the development and progression of AML. Clearly, it is essential to understand the mechanisms of apoptosis to establish a personalized, patient-specific approach in AML therapy. Therefore, this paper comprehensively reviews the current range of AML treatment approaches related to apoptosis and highlights other promising concepts such as neddylation.

**Abstract:**

More than 97% of patients with acute myeloid leukemia (AML) demonstrate genetic mutations leading to excessive proliferation combined with the evasion of regulated cell death (RCD). The most prominent and well-defined form of RCD is apoptosis, which serves as a defense mechanism against the emergence of cancer cells. Apoptosis is regulated in part by the BCL-2 family of pro- and anti-apoptotic proteins, whose balance can significantly determine cell survival. Apoptosis evasion plays a key role in tumorigenesis and drug resistance, and thus in the development and progression of AML. Research on the structural and biochemical aspects of apoptosis proteins and their regulators offers promise for new classes of targeted therapies and strategies for therapeutic intervention. This review provides a comprehensive overview of current AML treatment options related to the mechanism of apoptosis, particularly its mitochondrial pathway, and other promising concepts such as neddylation. It pays particular attention to clinically-relevant aspects of current and future AML treatment approaches, highlighting the molecular basis of individual therapies.

## 1. Introduction

Acute myeloid leukemia (AML) is a molecularly and clinically-heterogeneous disease characterized by uncontrolled clonal proliferation of aberrantly differentiated myeloid progenitor cells in the bone marrow. This differentiation, and the resulting genomic instability associated with numerous genetic alterations, including both mutational and epigenetic changes, results in the accumulation of abnormal myeloid cells and the suppression of regular hematopoiesis [1,2]. Such mutations are present in more than 97% of AML patients and result in unrestrained proliferation via the evasion of regulated cell death (RCD) [3].

RCD, or the cell suicide pathway, plays a critical role in organismal development and homeostasis. The most prominent and well-defined form of RCD is apoptosis, which serves as a defense mechanism against the emergence of cancer cells [4] and serves as a homeostatic mechanism to maintain cell populations in tissues [5,6].

Apoptosis dysregulation facilitates the development and progression of AML by permitting the continued survival of cells with activated oncogenes. However, much has been learned about the structural and biochemical details of apoptosis proteins and their upstream regulators, and this knowledge has opened possibilities for the design of new classes of personalized therapies. Indeed, several components of the apoptotic pathway in leukemic cells have already been highlighted as potential therapeutic targets [7].

Despite these advances, the treatment of AML remains a challenge. Hence, new therapeutic concepts are needed for treating acute leukemia in daily clinical practice. Such development should certainly include an individual, tailored approach to treatment that takes into account complex molecular diagnostics. Therefore, it is critical to have a thorough understanding of apoptosis and its mechanisms before effective strategies can be designed for improving AML treatment.

The aim of this review is to provide a comprehensive overview of current AML treatment approaches related to the mechanism of apoptosis, specifically its mitochondrial pathway, and other promising concepts such as neddylation. This review examines clinically-relevant aspects of current and future AML treatments, highlighting the molecular background of individual therapies.

## 2. Apoptosis

Apoptosis (from Ancient Greek “falling off”) is a physiological process of cellular suicide required for controlling tissue homeostasis, particularly in rapidly-renewing tissues such as hematopoietic tissue [8]. It is characterized by distinct morphological changes including nuclear condensation, cell shrinkage, membrane blebbing, DNA fragmentation, and the formation of apoptotic bodies [9]. This elimination occurs naturally in the course of organ development or following cellular stress [10]. However, such stress signals can be overcome by cancer cells overexpressing anti-apoptotic proteins, especially those of the BCL-2 (B-cell lymphoma-2) family [10]. The *BCL2* gene acts as an oncogene that prevents hemopoietic cell death [11]. It has been proposed that the blockage of apoptotic signaling promotes oncogenesis, and this has been demonstrated in several model systems [12,13].

## 3. Pathways of Apoptosis

Apoptosis can occur by the intrinsic (or mitochondrial) and extrinsic (or death receptor) pathways [14]. The intrinsic pathway serves to activate apoptosis as an appropriate response to various internal traumas, such as metabolic stress, hypoxia, checkpoint violations, growth factor withdrawal, activation of oncogenes, and irreparable genomic damage. The extrinsic pathway is distinctly triggered by the ligation of proapoptotic transmembrane death receptors from the tumor necrosis factor (TNF) family [15]. A common feature of apoptosis is the involvement of caspases, a family of intracellular cysteine proteases, which are present as inactive zymogens in all animal cells, but can be triggered to assume an active state [8,16]. Caspases can be broadly categorized based on their role in apoptosis (caspase-2, -3, -6, -7, -8, -9, and 10) then further subdivided into two groups: the initiator caspases (caspase-2, -8, -9, and -10) and the effector caspases (caspase-3, -6, and -7) [11].

## 4. The Mechanism of the Mitochondrial Apoptosis Pathway

The mitochondrial apoptosis pathway is the dominant form of cell death, leading to the death of over 60 billion cells each day [17]. The process is initiated by stress signals and is triggered by mitochondrial outer membrane permeabilization (MOMP), caused by *inter alia* BCL-2 family proteins.

MOMP leads to the release of apoptogenic proteins from the intermembrane space. It results in the secretion of various cell death modulators such as cytochrome c, apoptosis-inducing factor (AIF), endonuclease G (ENDOG), direct IAP-binding protein with low pI (DIABLO, also known as second mitochondria-derived activator of caspases, SMAC) or Omi/HtrA2, and BCL-2 family proteins; it also prevents mitochondrial ATP synthesis, inhibits the respiratory chain and increases reactive oxygen species (ROS) production [18]. These events promote the activation of the initiator caspase 9 and executioner caspases (caspases 3, 6, and 7) in order to destroy the cell [19]. A key element in the regulation and induction of intrinsic apoptosis is the BCL-2 protein family, whose pro- and anti-apoptotic members, and the balance between them, play significant roles in determining the fate of cells [17,20] (Figure 1).

## 5. BCL-2 Family

The BCL-2 family, which tightly regulates MOMP, consists of three subfamilies: the first is the pro-apoptotic BH3-only members (BIM [BCL-2-like protein 11], BID [BH3 Interacting Domain Death Agonist], PUMA [p53 upregulated modulator of apoptosis], NOXA [Phorbol-12-myristate-13-acetate-induced protein 1], HRK [Activator of apoptosis harakiri], BMF [BCL-2 modifying factor], and BAD [BCL-2 associated agonist of cell death]); the second is the pro-apoptotic effector molecules (BAX [BCL-2 associated X] and BAK [BCL-2 homologous antagonist killer]); and the third is the anti-apoptotic BCL-2 family proteins (BCL-2, BCL-xL [B-Cell Lymphoma-extra-large], BCL-W [BCL-2 like protein 2], MCL-1 [myeloid cell leukemia-1], A1 [BCL-2 related protein A1], and BCL-B [BCL-2 like protein 10]).

The receipt of an apoptotic stimulus upregulates the transcription of the pro-apoptotic BH3-only members. These proteins bind anti-apoptotic members of the BCL-2 family and inhibit their activity. Some direct activators (for example, BID and BIM) can also bind and activate the effectors BAK and BAX, which induces MOMP, resulting in the release of cytochrome c and SMAC from mitochondria and the subsequent activation of initiator caspases [21,22].

Another protein regulating apoptosis is the p53 tumor suppressor, known as the ‘guardian of the genome’. It induces cell-cycle arrest with DNA repair or apoptosis by activating the BCL-2 family. The activation of p53 triggers pro-apoptotic proteins such as PUMA, NOXA, BIM, and BAX to counteract MCL-1, thus overcoming MCL-1 mediated resistance at multiple levels. Moreover, p53 stimulates extrinsic pathway activation by the upregulation of death receptors [7,23,24] (Figure 1).

## 6. Evasion of Apoptosis in AML

The expression of programmed cell death genes is commonly dysregulated in hematological malignancies, promoting cell accumulation and creating favorable conditions for oncogene activation, genetic instability, and metastasis. Such defects enable the survival of genetically unstable cells, leading to the selection of progressively aggressive clones [25]. Multiple elements of the apoptotic pathway are known to be disturbed in AML, thus facilitating the evasion of apoptosis.

The most common pathway used by cancer cells to evade apoptosis is based on the upregulation of antiapoptotic BCL-2 proteins and the loss of BAX/BAK proteins [26]; indeed, *BCL-2* gene overexpression is present in over half of all cancers [18]. As many traditional anti-cancer drugs act on the BCL-2/BAX pathway, any disruption of this target may increase resistance to chemotherapy and radiotherapy by elevating the threshold needed for cell death [26].

*TP53* gene mutation prevents apoptotic pathway function [26]. Such mutations are observed in 5–10% of de novo AML and 30–40% of therapy-related cases and are considered important indicators of poor outcome [27,28]. In a study conducted on 500 AML patients, Hou et al. found *TP53* mutation to be associated with an inferior response rate (complete remission [CR] rate 28.6% vs. 80.2%, *p* < 0.0001) and shorter overall survival (OS) (median, 5 vs. 35 months, *p* < 0.001) compared to unmutated patients [29].

Unlike solid tumors, in most cases of non-complex karyotype de novo AML, the *TP53* locus is found to be the wild-type. Most *TP53* changes result in missense mutations, i.e., resulting in changes in the amino acid sequence of the DNA-binding domain (encoded by exons 5–8). These are more common in complex karyotypes, relapsed, and elderly AML patients, as well as therapy-related AML [28,30,31]. Where the mutation is absent, as noted in the majority of AML cases, it is possible that apoptosis is blocked by alternate mechanisms [32]. Intriguingly, patients with a *TP53* mutation have a lower-than-expected frequency of mutations in other myeloid-related genes, involving splicing, epigenetics, and signal transduction [28,33]. However, irrespective of *TP53* mutational status, many AML cases are characterized by p53 dysfunction, presumably through the alteration of p53-regulatory proteins, resulting in the disruption of apoptosis [7,34].

The wild-type *TP53* may be significantly inhibited by high levels of mouse double minute 2 homolog (MDM2) or inhibitor of apoptosis proteins (IAP); this presents an appealing targeted strategy for AML treatment [35].

IAPs consist of several proteins which inhibit apoptosis pathway at different levels, namely X-linked inhibitor of apoptosis (XIAP), cellular (cIAP1, cIAP2), neuronal (NIAP), testis-specific (Ts-IAP), Bir-ubiquitin conjugating enzyme (BRUCE), livin, and survivin [36]. IAPs take part in modulating autophagy, necroptosis, and immune regulation, and can inhibit both intrinsic and extrinsic apoptosis; they also stimulate cell survival and increase tumor growth and metastasis when present at elevated levels [37,38]. Moreover, IAP overexpression reduces CR rate in AML patients and increases chemoresistance in several types of cancer. In addition, low IAP expression has also been associated with longer OS in AML patients [36,39,40]. Lack of IAP expression is also associated with higher sensitivity to standard chemotherapy [7,41,42]. Several molecules antagonize IAP activity, with the most prominent ones being SMAC/DIABLO, Omi/HtrA2 (HTRA serine peptidase 2), and XAF-1 (XIAP-associated factor 1) [43,44].

## 7. An Overview of the Impact of Dysregulation in the Expression of Apoptosis Genes on AML

One of the malfunctions in AML is the overexpression of anti-apoptotic protein BCL-2, which increases the threshold level of proapoptotic signaling required for initiating apoptosis [26,45]. High *BCL-2* mRNA expression has been noted in approximately 65% of 119 de novo AML patients and was connected with a significantly worse three-year OS than in those with little or no expression (10% vs. 49%, respectively) [46,47]. A similar trend was observed by Campos et al. and Tóthová et al. [32,48].

Another study conducted on 198 AML patients by Kornblau et al. showed that high BCL-2 protein levels may result in a lowered median survival in the case of favorable/intermediate cytogenetics, but a higher survival in the case of poor cytogenetics [49]. However, Zhou et al. did not find that *BCL-2* overexpression had any negative effect on clinical outcomes, and it was significantly up-regulated in newly-diagnosed AML patients [50].

In addition, disturbances in apoptosis may be particularly characteristic of some AML subtypes. AML1-ETO fusion protein related to AML with t (8; 21) has been shown to trigger the transcription of *BCL-2* [27]. In addition, *BCL-2* expression is likely to be induced by transcription factor CCAAT/enhancer binding protein α, encoded by the *CEBPA* gene, which is mutated in about 10% of AML patients [30,51,52].

In a study conducted on 235 AML patients, Schaich et al. reported that neither *BCL-2* nor *BAX* nor *BCL-XL* expression had a significant influence on OS or on disease-free survival (DFS). However, enhanced *BCL-XL* expression was found in 38% of AML patients, and surprisingly, an elevated *BAX* mRNA expression was found in 74%. Moreover, *BCL-XL* overexpression was found to be an independent negative prognostic factor for response to induction therapy in patients with intermediate risk cytogenetics, and the *BAX* gene was found to be less frequently expressed in the high-risk group than in the intermediate risk group [53].

In addition, higher expression of *BAX* has been associated with a higher incidence of CR rate [54]. However, surprisingly, another study which assessed 232 samples from AML patients found that a higher *BAX* and *BAD* expression correlated with poor outcome [55].

*BCL-XL* upregulation has also been detected in patients with mutations in fms-like tyrosine kinase 3 (FLT3) and c-KIT, resulting in the activation of a signal transducer and activator of transcription (STAT) [56,57]. It also caused the constitutive activation of the phosphoinositide 3-kinase (PI3K)-AKT signaling pathway in myeloblasts, resulting in the eventual suppression of apoptosis via phosphorylation of pro-apoptotic BAD by inhibiting its interaction with BCL-2 [58]. In FLT3-internal tandem duplication (ITD) mutant AML cells, similar anti-apoptotic effects of BAD phosphorylation have been identified [46,59,60].

Another important member of the anti-apoptotic protein family is MCL1. It has prevented apoptosis by binding to the pro-apoptotic BCL-2 proteins and is thought to be vital for the development and maintenance of AML [5]. MCL-1 overexpression has been associated with tumorigenesis, poor prognosis, and drug resistance [61]. High MCL-1 expression has also been associated with relapse in AML patients [48,49]. This has been particularly evident in the presence of the FLT3-ITD mutation, which upregulated MCL1 through STAT5 activation to prolong the survival of leukemic stem cells [7,49,61]. Li et al. reported the presence of high *MCL-1* mRNA expression in 50% of the studied AML patients and noted that this was connected with a shorter OS than in patients with lower *MCL-1* expression [62]. In addition, *MCL-1*, *BCL-XL,* and *BAK* were found to be expressed at lower levels in the high-risk AML group based on cytogenetic/molecular characteristics [63].

Mutations in the *TP53* gene have been evaluated in AML many times [64], but little is known about the influence of abnormalities in its expression. It has been found that *TP53* gene expression may be lowered in nearly 90% of AML patients, indicating that gene expression levels may be more important in AML than gene mutation [65].

The *SMAC/DIABLO* gene has been shown to be expressed in nearly 90% of AML patients, and a high *SMAC/DIABLO* expression has also been associated with increased DFS and OS rate [66]. Pluta et al. reported that SMAC/DIABLO proteins were found in 98% of AML patients and that this high level is a good predictor of CR achievement and longer OS [67].

## 8. Targeting BCL-2 in AML

Apoptosis-based therapies for AML continue to be a significant objective of research. One current therapy is the BCL-2 inhibitor venetoclax (ABT-199), which unlike other targeted treatments in AML, like FLT3 or IDH1 inhibitors, can be used in a molecularly heterogeneous group of patients [68].

AML monotherapy with the BCL-2 inhibitor has not been entirely successful [69]. However, the results of a multicenter phase Ib study evaluating the efficacy and safety of venetoclax in combination with decitabine or azacitidine has offered hope [70]. The study was conducted on a population of 145 patients aged ≥ 65 years who were not eligible for intensive chemotherapy. The median OS for all patients was 17.5 months, with a median follow-up time of 15.1 months. Overall response rates (CR + CR with incomplete blood count recovery [CRi]+ partial remission [PR]) were similar in the venetoclax 400 mg + azacitidine group and the venetoclax 400 mg + decitabine group (76% and 71%, respectively) [70]. The findings indicate that the combination of venetoclax and decitabine/azacytidine demonstrated good tolerability and a favorable remission rate (CR + CRi: 67%) [70]. In addition, the combination therapy demonstrated beneficial effects in the presence of high-risk factors such as age ≥ 75 years, high cytogenetic risk, or the presence of secondary AML [70]. This was the first study to achieve such promising therapeutic success among AML patients not eligible for intensive treatment. It should be emphasized that achieving a CR and improving the general condition of the patient offers the possibility of requalification for other types of treatment with hematopoietic stem cell transplantation as the only curative option.

A multicenter phase Ib/II study evaluated the efficacy of venetoclax in combination with low-dose cytarabine (LDAC) in 82 patients aged ≥60 years ineligible for intensive chemotherapy [71]. In total, 49% of the patients were characterized by secondary AML with an extremely unfavorable prognosis. Of the patients, 54% achieved CR/CRi, of which 26% were CR and 28% achieved CRi. The median OS in the analyzed cohort was 10.1 months [71].

In the VIALE-A prospective, randomized phase III study (NCT02993523) [72], 431 patients with AML who were ineligible for intensive chemotherapy were randomly allocated to receive venetoclax with azacitidine (*n* = 286) or placebo with azacitidine (*n* = 145). The analysis revealed a significantly longer OS in the venetoclax + azacitidine group (14.7 months) compared to the venetoclax + placebo group (9.6 months) and a significantly higher rate of achieving CR/CRi (66.4% vs. 28.3%) [72]. The results confirm previous observations that venetoclax in combination with DNA hypomethylating agent (HMA) is an excellent option for first-line treatment in patients with AML who are not eligible for intensive chemotherapy [72].

The VIALE-C randomized phase III trial (NCT03069352) assessed whether venetoclax in combination with low-dose cytarabine (LDAC) improves OS in previously-untreated AML patients ineligible for intensive chemotherapy compared with LDAC plus placebo [73]. A total of 211 patients were randomised to either venetoclax 600 mg/d in combination with LDAC (*n* = 143) or to placebo + LDAC (*n* = 68). In total, 38% of participants had secondary AML and 20% had previously been treated with HMA. At the primary analysis, the study did not meet its primary endpoint of a statistically significant improvement in OS. However, in an additional 6-month follow-up, venetoclax + LDAC resulted in a longer OS than LDAC monotherapy (8.4 vs. 4.1 months). CR/CRi rates were 48% for combination treatment vs. 13% for LDAC monotherapy [73]. Hence, venetoclax with LDAC yielded clinically significant improvements in remission rate and OS compared to LDAC monotherapy, with an acceptable safety profile [73].

Data from prospective, randomized phase III trials, as well as results from earlier studies, confirm the effectiveness of venetoclax when used in combination with a hypomethylating agent or low-dose cytarabine as a backbone therapy. This is especially true for older patients with newly-diagnosed AML who are not candidates for intensive induction chemotherapy [74].

The possibility of combining venetoclax treatment with intensive chemotherapy is currently being investigated (NCT03709758). There are high hopes for this therapeutic option, especially for groups of patients with unfavorable cytogenetic-molecular risk and with relapsed or the initially refractory disease. A prospective single-center phase Ib/II study evaluating fludarabine, cytarabine, G-CSF, and idarubicin combined with venetoclax (FLAG-IDA + VEN) in patients with newly-diagnosed or relapsed/refractory (r/r) AML demonstrated a 98% overall response rate ([ORR]: CR + CR with partial hematologic recovery [CRh] + CRi + morphologic leukemia free state [MLFS] + PR) and an 89% composite CR (CRc + CRh + CRi) [75]. In a phase II trial investigating the activity and safety of 3 + 7 daunorubicin and cytarabine plus venetoclax (DAV) chemotherapy in adults with AML, a high CR + CRi rate (91%) was achieved after one cycle of DAV, with a high rate of measurable residual disease (MRD) negativity (97%), as well as manageable adverse effects [76].

Promising results may also be obtained in the ongoing VIALE-T study evaluating the effectiveness of venetoclax in combination with azacitidine versus the standard of care in AML patients as a maintenance after allogeneic hematopoietic stem cell transplantation (allo-HSCT) (NCT04161885). Although venetoclax is currently used most widely in combination with HMA or LDAC, it is hoped that therapy will soon be optimized by replacing azacitidine with molecularly targeted therapy (dual molecularly targeted therapy) or adding molecularly targeted therapy to the venetoclax-HMA backbone (triple therapy) [77].

There are several ongoing clinical trials in AML patients investigating various combinations of regimens containing venetoclax, as well as other drugs inhibiting the BCL-2 family. More detailed information is provided in Table 1.

## 9. In the Grip of Apoptosis—The Desired Future after Venetoclax

### 9.1. Targeting BCL-XL

Among the agents targeting the BCL-2 family, only venetoclax has FDA approval. There is a need to identify new alternatives targeting the intrinsic pathway of apoptosis to overcome resistance to venetoclax [20,57,78], which often occurs due to the upregulation of other anti-apoptotic proteins such as MCL and BCL-XL. In addition, venetoclax has limited utility for the treatment of solid tumors, which are generally not dependent on BCL-2 for survival; however, BCL-XL can be overexpressed in many solid tumors, as well as in a subset of leukemia cells, and its expression is highly correlated with chemoresistance, independent of *TP53* mutational status [79].

To find treatments that can eliminate resistance to standard BCL-2 inhibitors, possible new approaches involving the induction of alternative antiapoptotic BCL-2 family members, such as BCL-XL and MCL-1, are under investigation [56,58]. This has resulted in the discovery of promising anticancer drug candidates including ABT-263 (navitoclax), as well as several BCL-XL and MCL-1 monoselective inhibitors [79,80].

Navitoclax, which inhibits BCL-2/BCL-XL/BCL-W, has been extensively tested in lymphoid malignancies and myeloproliferative neoplasms; however, its applicability in AML has proven to be limited due to significant thrombocytopenia, since platelets solely depend on BCL-XL for survival [81,82]. Therefore, despite being a key cancer target, BCL-XL is currently without any safe or effective therapy [79,83]. NCT05222984 is a new study evaluating navitoclax with venetoclax and decitabine in treating patients with r/r AML after previous treatment with venetoclax.

Nevertheless, two more advanced molecules targeting BCL-XL (PROTAC DT2216) and BCL-XL/2 (PROTAC 753B) have recently appeared; these constitute a new emerging technology for the development of novel and safe targeted anticancer therapeutics. PROTACs are small molecules, superior to conventional small molecule inhibitors (SMIs), that utilize the ubiquitin proteasome system (UPS) to degrade proteins of interest (POI) [84]. What makes them unique is their improved target selectivity and ability to target “undruggable” and mutant proteins [85]. Upon binding to the POI, the PROTAC can recruit E3 ligase for POI ubiquitination, which is subjected to proteasome-mediated degradation [86].

DT2216 is a first-generation BCL-XL PROTAC that is more potent against BCL-XL-dependent tumor cells and demonstrates reduced platelet toxicity than navitoclax. DT2216 targets the von Hippel–Lindau (VHL) E3 ligase at BCL-XL for degradation; this provides an advantage over navitoclax by protecting platelets, where VHL is poorly expressed [84,87].

753B PROTAC, which causes BCL-XL/BCL-2 ubiquitination and selective degradation of cells expressing VHL, is an innovative approach to inducing apoptosis activation and eliminating chemotherapy-induced senescent leukemia cells [88]. It has been found that 753B effectively eliminates AML cells and enhances the efficacy of chemotherapy by targeting senescent cells; this study also confirmed recent findings that standard chemotherapy with cytosine arabinose (Ara-C) induced cellular senescence (SnCs) in AML cells, manifested by increased cell size, the induction of senescence-associated β-galactosidase activity, the upregulation of cell cycle regulator proteins (p16, p21, p53), and the expression of the senescence-associated secretory phenotype factors (IL-6, IL-18, IL-1β), leading to chemoresistance in AML [89]. 753B has been found to reverse chemotherapy-induced SnCs phenotype and induce death in senescent AML cells by increasing BCL-XL/BCL-2 expression, and to trigger time and dose-dependent BCL-XL degradation at concentrations lower than that of DT2216 [90,91]. 753B also showed potency in nine venetoclax-resistant samples, of which six demonstrated BCL-2 degradation [88,92].

Another drug targeting BCL-2/BCL-XL, which is currently under research in the NIMBLE study in patients with relapsed/refractory AML, is AZD0466 [93]. AZD0466 is a drug-dendrimer conjugate consisting of AZD4320 (BCL2/XL dual inhibitor) covalently conjugated to a pegylated poly-L-lysine dendrimer; the drug is gradually released by hydrolysis, allowing for more efficient delivery [94].

### 9.2. Targeting MCL-1

MCL-1 overexpression can be identified in patients with relapsed AML. It was also found to be a major factor in resistance to the dual BCL-2/BCL-XL inhibitors in AML-derived cell lines and blast cells from AML patients [95]. MCL-1 not only contributed to leukemic stem cell survival and MRD, but also facilitated AML progression. Contrary to the clinical success of venetoclax, MCL-1 inhibitor clinical trials have been more challenging due to MCL-1 demonstrating a less flexible binding site than BCL-2 or BCL-XL. However, S64315/MIK665 (NCT04629443, NCT03672695, NCT04702425), demonstrating activity against MCL-1, entered phase I/II clinical trials in AML, while AMG 176 (NCT02675452) and PRT1419 (NCT04543305, NCT05107856) have entered phase I. An interesting approach is to combine S64315/MIK665 with S65487 (VOB560); the latter is a selective BCL-2 inhibitor, which is also active against BCL-2 mutations such as G101V and D103Y, and is known to induce robust anti-tumor activity (NCT04702425).

Unfortunately, many trials on anti-MCL-1 agents have been halted due to cardiac toxicity, which may be caused by deficiencies in the respiratory chain and ATP production, accompanied by increased ROS production [96]. Some, like that evaluating AZD5991 (NCT03218683), have been put on hold to allow further evaluation of safety-related information. Nevertheless, the development of MCL-1 PROTACs raises hopes for reducing on-target cardiotoxicity [84,97].

Moreover, like BCL-XL, MCL-1 expression is also upregulated in both hematologic malignancies and some solid tumors, providing another advantage over BCL-2 inhibition [84,98].

A promising approach is based on the use of indirect MCL blockade using cyclin-dependent kinase 9 (CDK9) inhibitors. These have demonstrated preclinical activity in AML, which has been primarily attributed to the depletion of MCL-1 protein with subsequent apoptosis. Several clinical trials with CDK9 inhibitors like alvocidib (NCT03298984, NCT03441555, NCT01349972), dinaciclib (NCT03484520), CYC065 (NCT04017546) and AZD4573 (NCT03263637) have been underway to evaluate their effectiveness in combination with other therapeutics [99,100,101].

### 9.3. Targeting IAPs

IAP overexpression induces chemotherapeutic drug resistance in many cancer types. Exploring the apoptosis-inducing activity of IAP inhibitors in AML provides new opportunities to sensitize drug-resistant AML cells to other therapeutics. These antagonists have been partially tested in both pre-clinical and clinical studies, mainly in combination therapies [102,103,104].

One new drug is ASTX660 (a dual antagonist of both cIAP1/2 and XIAP). It is currently in a clinical trial in combination with ASTX727 (decitabine/cedazuridine, hypomethylating agent), in patients with relapsed/refractory AML (NCT04155580) [105].

Upregulation of the smallest IAP, survivin, correlates with cancer progression and poor prognosis in AML [106]. A unique strategy for downregulating survivin expression involves the use of antisense oligonucleotide (ASO) technology, investigated as LY2181308 in a phase II study (NCT00620321). In this trial, the molecule demonstrated a good safety profile and some clinical benefit in r/r AML patients when used in combination with cytarabine and idarubicin [107,108]. However, despite initially promising results, phase II of the study examining the second-generation bivalent SMAC mimetic birinapant (NCT01486784) in r/r AML patients was never initiated [109,110].

### 9.4. Targeting p53—Dependent Apoptosis Pathways

Mutations in the *TP53* gene result in impaired apoptosis due to innate resistance to conventional chemotherapy. The effect of HMA may appear initially satisfactory, but as with allo-HSCT in this group of patients, it does not provide a long-term response. Consequently, all patients with *TP53* mutations are a source of great interest in clinical trials using new therapeutic approaches [28,111].

Eprenetapopt (APR-246) is a novel, first-in-class, small molecule that selectively induces apoptosis in *TP53*-mutant cancer cells. APR-246 is spontaneously converted into methylene quinuclidinone, which covalently binds to cysteine residues in mutant p53, thus stabilizing and shifting the equilibrium toward a functional conformation. The restoration of wild-type *TP53* expression triggers cell death by inducing apoptosis in tumor cells [112]. Eprenetapopt can also induce p53-independent cell death in the course of ferroptosis, a recently described cell death process [113].

Two recent phase II studies (NCT03072043 and NCT03588078) have found that eprenetapopt in combination with azacitidine demonstrated a promising safety profile and response rates [112,114,115]. However, a phase III study (NCT03745716) has also compared eprenetapopt plus azacitidine with azacitidine alone in the treatment of patients with myelodysplastic syndrome (MDS), and the coexisting *TP53* mutation failed to meet its primary endpoint concerning CR rate.

Currently, there is an ongoing phase I trial evaluating a three-drug combination comprising azacitidine, APR-246, and venetoclax in *TP53* mutated treatment-naive AML patients (NCT04214860); in addition, another multi-center, open-label, phase II clinical trial is assessing APR-246 in combination with azacitidine as maintenance therapy after allo-HSCT for patients with *TP53* mutant AML or MDS (NCT03931291) [116,117,118].

Moreover, Maslah et al. found that the synergistic effects of APR-246 with azacitidine were mediated by downregulation of the FLT3 pathway in drug-treated cells [118]. Therefore, in these patients, especially those with FLT3 mutations and high blast rates, combining treatment with FLT3 inhibitors may be a valuable option [119].

Although *TP53* mutations are more common in some AML subgroups, p53 inactivation is more likely to be caused by the overexpression of its endogenous inhibitors, like mouse double minute 4 homolog (MDMX) or mouse double minute 2 homolog (MDM2), which frequently occur in *TP53* wild-type AML [120].

Both the amplification and overexpression of MDM2 and MDMX have been associated with low p53 protein levels and unfavorable clinical outcome in AML patients [7,121]. They have also been found to be a main driver of malignancy, mainly when accompanied by wild-type *TP53* status. Han et al. reported that MDMX protein and mRNA are overexpressed in up to 92% of AML [122].

The MDM2 and MDMX homologs are important essential negative regulators of p53, acting as its gatekeepers. They are known to bind together and act most effectively as a complex. MDM2 largely inhibits and degrades p53 through ubiquitination, while MDMX interacts with p53 mainly by inhibiting transactivation activity [119,123,124]. Therefore, inhibition of the p53-MDM2/MDMX interaction represents an attractive therapeutic target, allowing reactivating of p53 function.

Idasanutlin (RG7388) is a representative of second-generation selective MDM2 antagonists, or nutlins, which block the interaction between the MDM2 protein and p53, thus preventing the enzymatic degradation of p53 and causing the transcriptional restoration of p53, leading to apoptosis [124]. However, the effective action of MDM2 antagonists is significantly limited to *TP53* wild-type AML patients.

Several phase I studies have reported that idasanutlin is a potentially safe option with promising clinical activity. In a multicenter phase I/Ib trial (NCT01773408) in 122 *TP53* wild-type AML patients, idasanutlin yielded a 10.8% CR as a single drug and 32.2% in combination with cytarabine. The composite clinical remission (CRc: CR + CRp [CR with incomplete platelet recovery] + CRi) rate in patients with *TP53* mutations was only 4.0% (*n* = 1/25). Median MDM2 expression was higher in patients who responded to idasanutlin than in non-responders [125].

Idasanutlin + cytarabine (1 g/m^2^) was compared with placebo + cytarabine in r/r AML in the MIRROS randomized phase III trial (NCT02545283); however, further analysis had to be stopped due to a failure to achieve study endpoints. The same was noted for the NCT03850535 trial with idasanutlin + cytarabine + daunorubicin [126]. In addition, the results of the NCT02670044 study evaluating the effect of idasanutlin in combination with venetoclax are awaited.

Combination targeted therapies may be particularly desirable, where the additional use of venetoclax has been shown to delay the development of resistance to MDM2 inhibitors. A phase Ib study (NCT02670044) found the combination of venetoclax + idasanutlin to demonstrate encouraging safety and efficacy in elderly patients with r/r AML who were ineligible for intensive chemotherapy. The anti-leukemic response rate, defined as CR + CRp, CRi + PR, and MLFS, was 41% with a median OS of 4.4 months [119,127].

Although MDM2 inhibitors have now advanced to clinical trials, those associated with MDMX have not yet been subjected to extensive clinical testing. The resistance and toxicity associated with MDM2 inhibitor treatment has driven the development of new therapies such as dual MDMX/MDM2 directed agents, which offer a more complete activity against p53 [128].

ALRN-6924 is a stabilized peptide and first-in-class MDM2/MDMX dual inhibitor, designed to activate p53. It demonstrates antileukemic activity through activation of p53-dependent transcription and induction of apoptosis [129]. In preclinical studies, ALRN-6924 was observed to inhibit the proliferation of AML cell lines and primary human AML cells, either alone or in combination with cytarabine, achieving CR in approximately 40% of mice in xenotransplantation studies [130]. Currently, ALRN-6924 is being tested in an early phase clinical trial involving patients with r/r AML and MDS [130,131].

Interestingly, ALRN-6924 does not induce deep levels of apoptosis in healthy tissues, even at the highest doses tested; however, it can induce reversible cell cycle arrest in properly proliferating cells. This concept, called cyclotherapy, offers the chance to selectively protect tissues, such as bone marrow, from cytotoxic chemotherapeutics, thus enabling cancer treatment with fewer side effects [128,132].

Novel anticancer therapies that reactivate p53 are a promising option; however, they require further research. It is likely that new therapeutic combinations are needed to truly improve long-term outcome and to have a reasonable possibility of introducing this group of drugs into routine clinical practice [28,119].

More detailed information on the results of studies involving the molecules described in the text is summarized in Table 2. Additionally, a summary of ongoing clinical trials has been provided in Table 3.

## 10. Ferroptosis—A New Type of Regulated Cell Death

Ferroptosis is an iron-dependent form of RCD. There is growing evidence that the boundary between ferroptosis and other types of RCDs appears to be blurred. This process is triggered by impaired redox and antioxidant mechanisms and is primarily featured by changes in iron homeostasis, iron-dependent lipid peroxidation, and accumulation of glutamate toxicity, which can be specifically reversed by ferroptosis inducers or inhibitors. Ferroptosis, similar to apoptosis, can be induced by both endogenous and exogenous pathways. The endogenous pathway is activated primarily by reducing the expression or activity of intracellular antioxidant enzymes. Meanwhile, the exogenous pathway is triggered by inhibition of cell membrane transporters (e.g., the cysteine/glutamate reverse transporter) or by induction of transferrin and ferroportin [133].

Ferroptosis is mediated by intercellular iron overload. After conjugation with transferrin, extracellular ferric iron (Fe3+) is imported into cells through the membrane protein transferrin receptor 1 (TfR1). As a result of the ferroptosis mechanism, cells lose the integrity of the plasma membrane, the cytoplasm becomes swollen, the mitochondria become smaller, the outer mitochondrial membrane breaks down, and the membrane density increases [134].

Cancer cells have a stronger need for iron compared to non-malignant cells. In recent years, the possibility has been presented that cancer cells that are resistant to conventional therapies may be particularly susceptible to ferroptosis. Moreover, ferroptosis has recently been shown to be associated with cancer immunotherapy, where T cells and INFγ promote tumor cell sensitivity to this type of RCD. It has now been established that AML is sensitive to ferroptosis-promoting compounds. Current research on the role of ferroptosis in AML is based on cell lines and generally involves the development of new therapeutic reagents that produce interference to a particular link in the mechanism of ferroptosis [135].

## 11. Future Directions in Apoptosis-Related Neddylation

Another aspect worth attention in targeted treatment connected with apoptosis is the inhibition of neddylation. Neddylation is a post-translational modification closely related to ubiquitination and apoptosis which conjugates a ubiquitin-like protein NEDD8 to substrate proteins. The conjugation of NEDD8 to its substrates is catalyzed by NEDD8-activating enzyme E1 (NAE), two NEDD8-conjugating enzyme E2s (UBC12/UBE2M and UBE2F), and several substrate-specific NEDD8-E3 ligases [136]. The best-characterized substrate of the NEDD8 pathway is the cullin family of proteins, Cul-1–3, 4A, 4B, 5, 7, and 9, also known as PARC (p53-associated Parkin-like Cytoplasmic Protein), which provide support for ubiquitin ligases (E3). Other NEDD8 pathway substrates include p53, p73, EGFR, caspases/IAPs/RIP1, pVHL, and Parkin/PINK1 [137] (Figure 2).

Neddylation has been associated with multiple cellular processes, including cell cycle progression, metabolism, immunity, and carcinogenesis. Moreover, it contributes to tumorigenesis and resistance to apoptosis by promoting ubiquitin/proteasome-dependent degradation of proteins involved in inhibiting cell growth. In addition, as neddylation upregulation is found in many human cancers, the NEDD8 pathway may be considered a novel therapeutic target [137,138].

## 12. Targeting Neddylation in AML

### 12.1. Pevonedistat

Pevonedistat (TAK-924/MLN4924) is an inhibitor of the NEDD8 activating enzyme (NAE), which suppresses neddylation and induces cell cycle arrest and apoptosis [139]. Moreover, it increases the accumulation of the pro-apoptotic protein NOXA, as well as the upregulation of PUMA, BIK, and BIM [140,141,142]. However, the phase III PANTHER study (NCT03268954) of pevonedistat plus azacitidine (*n* = 227) vs azacitidine (*n* = 227) in patients with newly diagnosed higher-risk MDS, higher-risk chronic myelomonocytic leukemia, or AML with 20–30% blasts, failed to reach pre-defined statistical significance for the primary endpoint of event-free survival (EFS) [143]. More detailed information on clinical trials concerning the neddylation pathway in AML is provided in Table 4.

### 12.2. TAS4464

The most potent and selective NAE inhibitor is TAS4464, which has demonstrated widespread antiproliferative activity against AML cell lines. It activates both the intrinsic and extrinsic apoptotic pathways and triggers caspases (especially caspase-8 and caspase-9) by increasing the NOXA level and decreasing the antiapoptotic factor c-FLIP. TAS4464 demonstrates a long-acting NAE inhibition effect in hematological malignancies. Unfortunately, its trial (NCT02978235) was terminated before Phase II due to cases of drug-induced liver injury [136].

## 13. Conclusions

Our knowledge of the mechanisms of apoptosis regulation continues to grow and this translates into new therapeutic possibilities. Despite this, not all new drugs in phase II and III clinical trials show satisfactory efficacy with acceptable toxicity. Nevertheless, the introduction of new personalized targeted therapies may effectively translate into higher response rates and longer survival among AML patients.

## Figures and Tables

**Figure 1 cancers-14-04995-f001:**
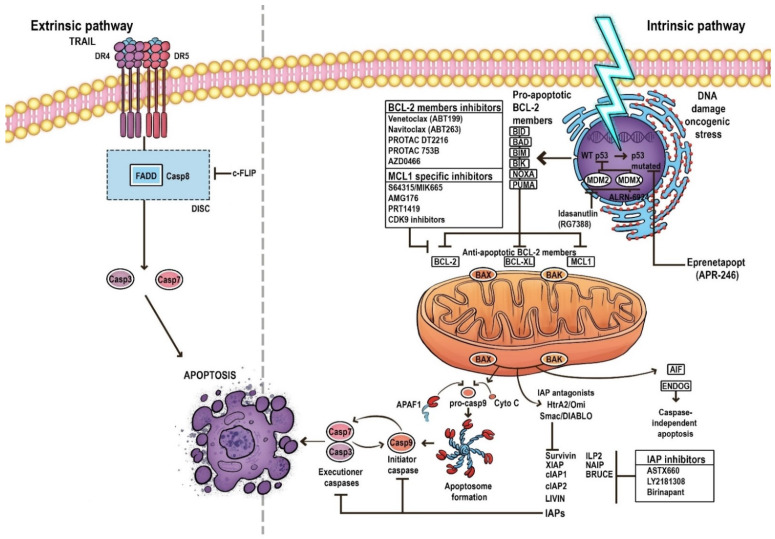
The pathways of apoptosis. Arrows represent activation and T bars represent inhibition.

**Figure 2 cancers-14-04995-f002:**
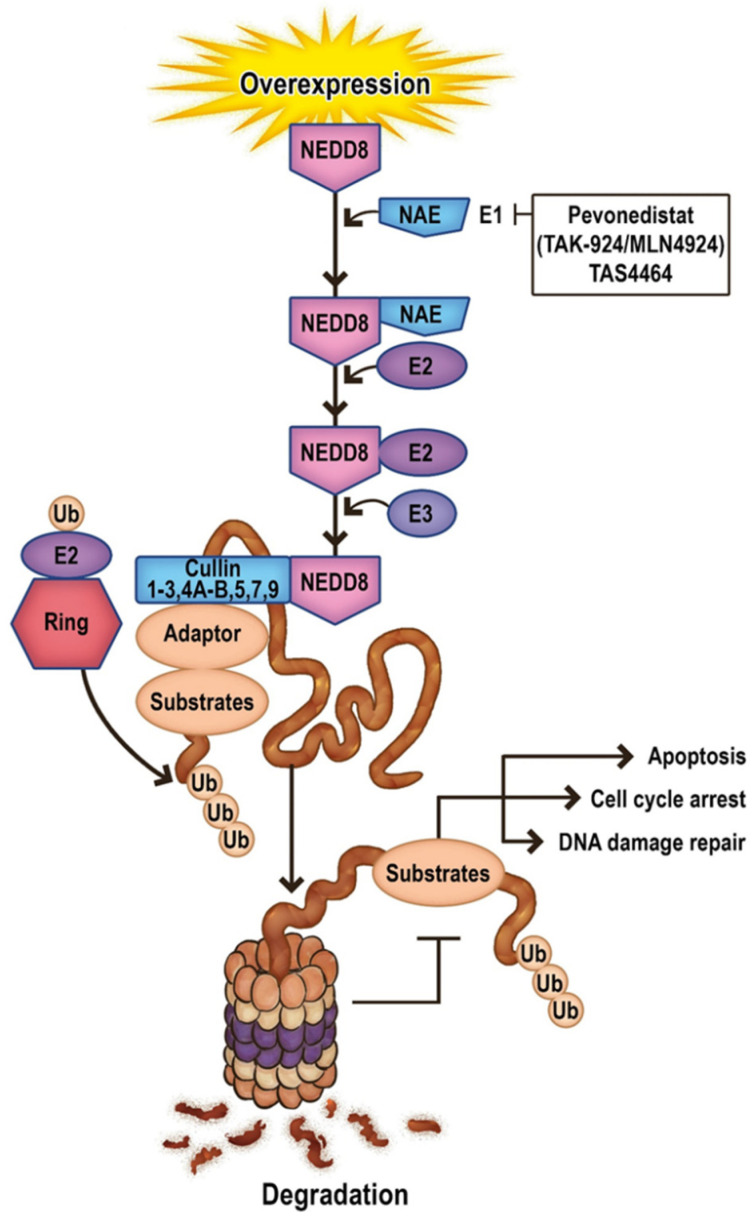
Schematic representation of the main steps of the neddylation pathway. Arrows represent activation and T bars represent inhibition.

**Table 1 cancers-14-04995-t001:** List of ongoing clinical trials in phase II-III in AML testing venetoclax (ABT-199).

Ongoing Trial	Intervention/Treatment	Clinical Trial Phase	Serial NCT Number	Current Status
CC-90011 given concurrently with Venetoclax and Azacitidine in r/r AML and treatment-naive subjects with AML who are not eligible for intensive induction chemotherapy.	CC-90011, Venetoclax, Azacitidine	Phase I/II	NCT04748848	Completed
Milademetan tosylate with cytarabine with or without ventoclax in treating participants with r/r AML.	Cytarabine, Milademetan Tosylate, Venetoclax	Phase I/II	NCT03634228	Completed
Low-dose cytarabine or azacitidine plus venetoclax and quizartinib in newly diagnosed AML patients aged equal or more than 60 years old.	Azacitidine, Venetoclax, Quizartinib, Cytarabine	Phase I/II	NCT04687761	Recruiting
CA-4948 monotherapy and in combination with azacitidine or venetoclax in AML patients.	Emavusertib, Azacitidine, Venetoclax	Phase I/IIa	NCT04278768	Active, not recruiting
Anti-OX40 antibody PF-04518600 (OX40) with or without venetoclax, avelumab, glasdegib, gemtuzumab ozogamicin, and azacitidine in treating r/r AML.	PF-04518600, Avelumab, Azacitidine, Gemtuzumab Ozogamicin, Glasdegib, Glasdegib Maleate, Venetoclax	Phase Ib/II	NCT03390296	Active, not recruiting
VEN-OM assessing venetoclax with escalating doses of omacetaxine in patients with r/r AML.	Omacetaxine, Venetoclax	Phase Ib/II	NCT04926285	Recruiting
ENAVEN-AML assessing enasidenib in combination with venetoclax in patients with IDH2-mutated r/r AML.	Enasidenib, Venetoclax	Phase Ib/II	NCT04092179	Recruiting
Omacetaxine with venetoclax in treating patients with r/r AML and have a genetic change RUNX1.	Omacetaxine Mepesuccinate, Venetoclax	Phase Ib/II	NCT04874194	Recruiting
IMGN632 with azacitidine and/or venetoclax in patients with relapsed and frontline CD123-positive AML, and antileukemia activity of IMGN632 when administered as monotherapy in patients with MRD+ AML after frontline treatment.	Azacitidine, IMGN632, Venetoclax	Phase Ib/II	NCT04086264	Recruiting
Decitabine/cedazuridine (ASTX727) and venetoclax in combination with ivosidenib or enasidenib in r/r AML.	Decitabine, Cedazuridine, Enasidenib, Ivosidenib, Venetoclax	Phase Ib/II	NCT04774393	Recruiting
Venetoclax in combination with quizartinib in r/r FLT3 (+) AML.	Quizartinib, Venetoclax	Phase Ib/II	NCT03735875	Active, not recruiting
Venetoclax with ivosidenib with or without azacitidine, in treating patients with IDH1-mutated AML.	Azacitidine, Ivosidenib, Venetoclax	Phase Ib/II	NCT03471260	Recruiting
Quizartinib, decitabine, and venetoclax in treating participants with acute myeloid leukemia that is untreated or has come back (relapsed).	Decitabine, Quizartinib, Venetoclax	Phase I/II	NCT03661307	Recruiting
DS-1594b with or without azacitidine, venetoclax, or mini-HCVD in treating patients with r/r AML.	DS-1594b, Azacitidine, Venetoclax, Mini-HCVD	Phase I/II	NCT04752163	Active, not recruiting
Gilteritinib with azacitidine and venetoclax in treating patients with FLT3-mutation positive r/r acute myeloid leukemia.	Azacitidine, Gilteritinib, Venetoclax	Phase I/II	NCT04140487	Recruiting
Venetoclax with azacitidine and pevonedistat in treating patients with newly diagnosed AML.	Azacitidine, Pevonedistat, Venetoclax	Phase I/II	NCT03862157	Active, not recruiting
Venetoclax and lintuzumab-ac225 in patients with with CD33 positive r/r AML.	Lintuzumab-Ac225, Venetoclax, Spironolactone	Phase I/II	NCT03867682	Recruiting
RELAX assessing venetoclax in combination with increasing cytarabine doses plus mitoxantrone in r/r AML.	Venetoclax, Cytarabine, Mitoxantron	Phase I/II	NCT04330820	Recruiting
Gilteritinib given together with ASTX727 and venetoclax in treating patients with FLT3-mutated AML that is newly diagnosed or r/r.	Decitabine, Cedazuridine, Gilteritinib, Venetoclax	Phase I/II	NCT05010122	Recruiting
Liposome-encapsulated daunorubicin-cytarabine (CPX-351) and venetoclax in r/r or untreated AML patients.	CPX-351, Venetoclax	Phase II	NCT03629171	Recruiting
Venetoclax, cladribine, low dose cytarabine, and azacitidine in treating patients with AML that has previously not been treated.	Azacitidine, Cladribine, Cytarabine, Venetoclax	Phase II	NCT03586609	Recruiting
AML secondary to myeloproliferative neoplasms unfit for intensive chemotherapy investigating a treatment combination including decitabine and venetoclax.	Venetoclax, Decitabine	Phase II	NCT04763928	Recruiting
Azacitidine, venetoclax, and trametinib in r/r AML.	Azacitidine, Trametinib, Venetoclax	Phase II	NCT04487106	Recruiting
Venetoclax together with busulfan, cladribine, and fludarabine in treating patients with high-risk acute myeloid leukemia who are undergoing stem cell transplant.	Busulfan, Cladribine, Fludarabine Phosphate, HSCT, Thiotepa, Venetoclax	Phase II	NCT04708054	Recruiting
STIMULUS-AML1 assessing the efficacy of MBG453 in combination with the HMA azacitidine and venetoclax.	MBG453, Venetoclax, Azacitidine	Phase II	NCT04150029	Active, not recruiting
(V-FIRST) assessing the efficacy of venetoclax in combination with fludarabine, cyratabine, and idarubicine in induction for AML patients with poor prognosis.	Venetoclax, Fludarabine, Cyratabine, Idarubicine	Phase II	NCT03455504	Recruiting
Decitabine, venetoclax, and ponatinib for the treatment of Philadelphia chromosome-positive AML.	Decitabine, Ponatinib, Venetoclax	Phase II	NCT04188405	Recruiting
Venetoclax plus azacitidine in newly diagnosed AML in patients who cannot receive intensive chemotherapy.	Low-dose Venetoclax, Azacitidine	Phase II	NCT05048615	Recruiting
Venetoclax and ASTX727 in relapsed/refractory (r/r) AML or unfit patients and the same combination for higher-risk AML patients without FLT3 (NCT04817241).	Decitabine, Cedazuridine, Venetoclax	Phase II	NCT04746235	Recruiting
Azacytidine + venetoclax versus conventional cytotoxic chemotherapy in induction-eligible AML patients.	Cytarabine, Idarubicin, Daunorubicin, Venetoclax, Azacitidine	Phase II	NCT04801797	Recruiting
Azacitidine and venetoclax with or without pembrolizumab in treating older patients with newly diagnosed acute myeloid leukemia who are ineligible or who refuse intensive chemotherapy.	Azacitidine, Pembrolizumab, Venetoclax	Phase II	NCT04284787	Recruiting
Venetoclax and sequential busulfan, cladribine, and fludarabine phosphate before donor stem cell transplant in treating patients with AML.	Busulfan, Cladribine, Fludarabine, Venetoclax	Phase II	NCT02250937	Active, not recruiting
Venetoclax and azacitidine for non-elderly adult patients with acute myeloid leukemia.	Venetoclax, Azacitidine	Phase II	NCT03573024	Recruiting
BP1001 (a liposomal Grb2 antisense oligonucleotide) in combination with venetoclax plus decitabine in patients with AML who are ineligible for intensive induction therapy.	BP1001, Ventoclax, Decitabine	Phase IIa	NCT02781883	Recruiting
Therapeutic efficacy of (venetoclax and decitabine) versus conventional “7 + 3” chemotherapy in induction young patients with AML.	Venetoclax, Decitabine, Cytarabine, Idarubicin	Phase III	NCT05177731	Recruiting
ENHANCE-3 study comparing the efficacy of magrolimab + venetoclax + azacitidine versus placebo + venetoclax + azacitidine in unfit de novo AML.	Magrolimab, Venetoclax, Azacitidine	Phase III	NCT05079230	Recruiting
ENHANCE-2 comparing the efficacy of magrolimab + azacitidine versus venetoclax + azacitidine in adults with previously untreated *TP53* mutant acute myeloid leukemia (AML) who are appropriate for non-intensive therapy as measured by overall survival (OS).	Magrolimab, Venetoclax, Azacitidine, Cytarabine, Daunorubicin, Idarubicin	Phase III	NCT04778397	Recruiting
VIALE-T assessing the efficacy of venetoclax in combination with azacitidine to improve Relapse Free Survival (RFS) in AML participants compared to Best Supportive Care (BSC) when given as maintenance therapy following allo-HSCT.	Venetoclax, Azacitidine	Phase III	NCT04161885	Recruiting

Abbreviations: AML, acute myeloid leukemia; MRD, measurable residual disease; FLT3, fms-like tyrosine kinase 3; mini-HCVD, hyperfractionated cyclophosphamide, vincristine, doxorubicin, and dexamethasone; HMA, DNA hypomethylating agent.

**Table 2 cancers-14-04995-t002:** Characteristics of apoptosis-targeting drug trials in AML.

Author; Year	Target Disease	Intervention Arm	Control Arm	Number of Patients	Important Findings
**BCL2 inhibitor (Venetoclax)**					
Konopleva et al. 2016	High-risk r/r AML or patients unfit for intensive chemotherapy	Ven	-	32	mOS—4.7 months
					CR/CRi—19%
					6-month leukemia-free survival rate was 10%
DiNardo et al. 2019	Naïve AML or patients unfit for intensive chemotherapy	Ven + HMA (AZA/DEC)	-	145	mOS—17.5 months
					CR/CRi—67%
					median duration of CR + CRi—11.3 months
Wei et al. 2019	Naïve AML or patients unfit for intensive chemotherapy	Ven + LDAC	-	82	mOS—10.1 months
					CR/CRi- 54%
DiNardo et al. 2020 VIALE-A	Naïve AML, ineligible for intensive chemotherapy	Ven + AZA	Placebo + AZA	431	mOS—14.7 months (ven) vs. 9.6 months (placebo)
					CR—36.7% (ven) vs. 17.9% (placebo)
					CR/CRi—66.4% (ven) vs. 28.3% (placebo)
Wei et al. 2021 VIALE-C	Naïve AML, ineligible for intensive chemotherapy	Ven + LDAC	Placebo + LDAC	211	Median OS—8.4 months (ven) vs. 4.1 months (placebo)
					CR/CRi- 48.3% (ven) vs. 13.2% (placebo)
					median event-free survival was 4.9 (ven) vs. 2.1 months (placebo)
**IAP Antagonists**					
Schimmer et al. 2011	r/r AML	AEG35156 + high-dose cytarabine and idarubicin	High-dose Ara-C + Idarubicin	40	CR/CRp—41% (intervention) vs. 69% (control)
					AEG35156 + reinduction chemotherapy did not improve rates of remission
Erba et al. 2013	r/r AML	LY2181308 alone or with Idarubicin + Ara-C	-	24	4/16 patients—morphologic complete response,
					1/16—an incomplete response,
					4/16—cytoreductions
					2/16—SD
Frey et al. 2014	r/r AML or high risk MDS refractory to HMA	Birinapant	-	20	Stable disease at 1, 3, and 6 months in some patients; one patient had a decline in their bone marrow blast count from 60% to 10%
**CDK9 inhibitors—Indirect MCL1 Inhibitors**					
Zeidner et al. 2021	Naïve AML ≤ 65 years	Alvocidib	-	32	ORR—75%
		Cytarabine			CR—69%
		Daunorubicin			Overall CR rates were 89%, 71%, and 56% for favorable-, intermediate-, and adverse-risk patients by ELN classification
Zeidner et al. 2015	Naïve AML with intermediate/adverse-risk cytogenetics	Alvocidib, Cytarabine, Mitoxantrone	Cytarabine	165	FLAM led to higher complete response rates than 7 + 3 alone (70% vs. 46%) without an increase in toxicity
		(FLAM)	+Daunorubicin		no significant differences in mOS
**Targeting p53**					
**1. The restoration of wild-type *TP53***					
Sallman et al. 2021	*TP53*-mutant AML with 20–30% marrow blasts or MDS	Eprenetapopt (APR-246) + AZA	-	55 (*n* = 11 AML)	overall response rate and CR rate for AML was 64% and 36%
					mOS—10.8 months
NCT03072043					
Cluzeau et al.	*TP53*-mutant AML or MDS	Eprenetapopt (APR-246) + AZA	-	53 (*n* = 18 AML)	In AML, the ORR was 33% including 17% CR
2021					72% achieved TP53 NGS negativity
					mOS—13.9 and 3.0 months in AML with less than and more than 30% marrow blasts, respectively
NCT03588078					
**2. MDM2 Inhibitors**					
Yee et al. 2021	r/r AML, s-AML, t-AML, or naïve AML, ineligible for intensive chemotherapy	Idasanutlin (RG-7388) + Ara-C	Idasanutlin	122	CRc—18.9 % with Idasa and 35.6 % with Idasa-Ara-C.
					CR—10.8 % and 32.2 % in those receiving Idasa and Idasa-Ara-C, respectively
Daver et al. 2019	r/r AML, ineligible for intensive chemotherapy	Idasanutlin + VEN	-	49	Anti-leukemic response rate—41%
					mOS—4.4 and 5.7 months in the VEN cohort

Abbreviations: AML, acute myeloid leukemia; Ven, venetoclax; mOS, median overall survival; CR/CRi, complete remission/CR with incomplete blood count recovery HMA, DNA hypomethylating agent; AZA, azacytidine; DEC, decitabine; LDAC, low-dose cytarabine; CR/CRp, CR/CR with incomplete platelet recovery; SD, stable disease; MDS, myelodysplastic syndrome; ORR, overall response rate; FLAM, fludarabine, cytarabine, and mitoxantrone; NGS, next-generation sequencing; Idasa, idasanutlin; s-AML, secondary AML; t-AML, therapy-related AML; Ara-C, cytosine arabinose.

**Table 3 cancers-14-04995-t003:** Ongoing clinical trials in AML involving molecules affecting apoptosis, as a single agent or in combination.

Agents and Ongoing Trials	Target Disease	Intervention Arm	Control Arm	Estimated Enrollment	Design; Clinical Trial Phase	Serial NCT Number	Current Status
**BCL-2, BCL-XL, BCL-W inhibitors**							
Navitoclax; BCL-2/BCL-XL/BCL-W inhibitor;	r/r AML Previously Treated with Venetoclax	Navitoclax, Venetoclax, Decitabine	-	24	Phase Ib, Open-Label study	NCT05222984	Recruiting
AZD0466 BCL-2/BCL-XL inhibitor NIMBLE	r/r AML	AZD0466	-	141	Phase I/II, Open-Label study	NCT04865419	Recruiting
**Direct MCL1 Inhibitors**							
S64315/MIK665	r/r AML, ineligible for intensive chemotherapy	S64315 + AZA	-	180	Phase I/II, Open-Label study	NCT04629443	Recruiting
S64315/MIK665	r/r AML, ineligible for intensive chemotherapy	S64315 + VEN	-	40	Phase Ib, Open-Label study	NCT03672695	Recruiting
S64315/MIK665	r/r AML, ineligible for intensive chemotherapy	S64315+ S65487 (VOB560)	-	170	Phase Ib, Open-Label study	NCT04702425	Recruiting
AMG176	r/r AML	AMG176 ± AZA	-	175	Phase I, Open-Label study	NCT02675452	Recruiting
PRT1419	r/r AML	PRT1419	-	36	Phase I, Open-Label study	NCT04543305	Active, not recruiting
PRT1419	r/r AML	PRT1419	-	30	Phase I, Open-Label study	NCT05107856	Recruiting
AZD5991	r/r AML, ineligible for intensive chemotherapy	AZD5991 ± VEN	-	144	Phase I/II, Open-Label study	NCT03218683	Terminated
**CDK9 inhibitors—Indirect MCL1 Inhibitors**							
Alvocidib	r/r AML	Alvocidib + VEN	-	36	Phase Ib, Open-Label study	NCT03441555	Completed
Dinaciclib/MK7965	r/r AML	Dinaciclib (MK7965) + VEN	-	48	Phase Ib, Open-Label study	NCT03484520	Active, not recruiting
CYC065	r/r AML	CYC065 + VEN	-	25	Phase I, Open-Label study	NCT04017546	Active, not recruiting
AZD4573	r/r AML	AZD4573	-	44	Phase I, Open-Label study	NCT03263637	Completed
IAP Antagonists							
ASTX660	r/r AML	ASTX660 ± ASTX727	-	68	Phase I, Open-Label study	NCT04155580	Terminated
**Targeting p53**							
**1. The restoration of wild-type *TP53***							
Eprenetapopt/APR-246	*TP53*-mutant AML	Eprenetapopt (APR-246) + AZA + VEN	-	51	Phase I, Open-Label study	NCT04214860	Completed
Eprenetapopt/APR-246	*TP53*-mutant AML after allo-HSCT	Eprenetapopt (APR-246) + AZA	-	33	Phase II, Open-Label study	NCT03931291	Completed
**2. MDM2 Inhibitors**							
Idasanutlin	Newly diagnosed AML	Idasanutlin+ Cytarabine + Daunorubicin	-	24	Phase Ib/II, Open-Label Study	NCT03850535	Terminated
Idasanutlin	r/r AML, ineligible for intensive chemotherapy	Idasanutlin + Cobimetinib + Venetoclax	-	88	Phase Ib, Open-Label Study	NCT02670044	Completed
Idasanutlin	r/r AML	Idasanutlin + Ara-C	Placebo + Ara-C	447	Phase III, Double-Blind, Randomized Study	NCT02545283	Terminated
**3. MDM2/MDMX inhibitors**							
ALRN-6924	r/r AML	ALRN-6924 ± Ara-C	-	55	Phase I/Ib Open-Label Study	NCT02909972	Completed

Abbreviations: r/r AML, relapsed/refractory AML; AZA, azacitidine; Ven, venetoclax; allo-HSCT, allogeneic hematopoietic stem cell transplantation; Ara-C, cytosine arabinose.

**Table 4 cancers-14-04995-t004:** Current ongoing clinical trials in AML involving molecules affecting neddylation.

Targeting Neddylation (NAE Inhibitors)							
Agents and Ongoing Trials	Target Disease	Intervention Arm	Control Arm	Estimated Enrollment	Design; Clinical Trial Phase	Serial NCT Number	Current Status
Pevonedistat/TAK-924/MLN4924	AML, ineligible for intensive chemotherapy	Pevonedistat + VEN + AZA	VEN + AZA	164	Phase II, Open-Label study	NCT04266795	Active, not recruiting
Pevonedistat/TAK-924/MLN4924	r/r AML	Pevonedistat + Belinostat	-	30	Phase I, Open-Label study	NCT03772925	Recruiting
Pevonedistat/TAK-924/MLN4924	Naïve high-risk AML with at least 1: adverse genetic features/t-AML/AML with antecedent MDS/≥55 years and considered fit for chemotherapy/AML with MDS-related changes	Pevonedistat + Ara-C + Idarubicin	-	53	Phase Ib/II, Open-Label study	NCT03330821	Active, not recruiting
Pevonedistat/TAK-924/MLN4924	r/r AML or r/r MDS	Pevonedistat + AZA + Ara-C + Fludarabine Phosphate + Methotrexate	-	12	Phase I, Open-Label study	NCT03813147	Active, not recruiting
Pevonedistat/TA K-924/MLN4924	Newly diagnosed s-AML or newly diagnosed CMML/MDS or post-HMA failure CMML/MDS	Pevonedistat +VEN + AZA	-	40	Phase I/II, Open-Label study	NCT03862157	Active, not recruiting
Pevonedistat/TAK-924/MLN4924	Newly diagnosed AML not eligible for intensive chemotherapy	Pevonedistat + AZA	AZA	466	Phase III	NCT04090736	Active, not recruiting
Pevonedistat/TAK-924/MLN4924	r/r AML (dose-escalation phase), newly diagnosed AML or r/r AML (expansion phase)	Pevonedistat +VEN + HMA	-	24	Phase I	NCT04172844	Active, not recruiting
Pevonedistat/TAK-924/MLN4924	AML or MDS after first CR with intensive chemotherapy or CR after allo-HSCT with MRD	Pevonedistat + AZA	AZA	102	Phase II	NCT04712942	Active, not recruiting

Abbreviations: Ven, venetoclax; AZA, azacytidine; r/r AML, relapsed/refractory AML; t-AML, therapy-related AML; MDS, myelodysplastic syndrome; Ara-C, cytosine arabinose; s-AML, secondary AML; CMML, chronic myelomonotic leukemia; HMA, hypomethylating agent; CR, complete remission; allo-HSCT, allogeneic hematopoietic stem cell transplantation; MRD, measurable residual disease.

## Data Availability

Not applicable.

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
