# Peer review of "Targeting Apoptosis in AML: Where Do We Stand?"

_cancers, 2022, doi:10.3390/cancers14204995_

Round 1

Reviewer 1 Report

The Authors have done nice work by putting together the mechanisms of apoptosis and targeting it for new treatment approaches in AML.

The authors can make this review better by

1. Adding few more latest references.

2. Doing final English language check and editing.

3. Follow the Journal"s recommendations in compiling the manuscript and references before publishing.

mechanisms of apoptosis to create a personalized, patient-specific approach in AML therapy. Therefore, this paper comprehensively reviews the current range of AML treatment approaches related to apoptosis, and highlights other promising concepts such as neddy

Author Response

Lodz 30th Aug 2022

Dear Reviewer,
Please find attached the manuscript entitled “Targeting apoptosis in AML: where do we stand?” which we are submitting for consideration for publication as the review article. 
We have introduced your suggestions, and have altered the manuscript 
accordingly. We feel that the revised manuscript is now much improved and hope that it will now be accepted for publication.

Here are the relevant changes:

1. Adding few more latest references.
- In our review, we have added some recent references, mainly those on the role of genetics, regulated cell death, and the importance of neddylation in AML.

2. Doing final English language check and editing.
- We have performed linguistic as well as editorial corrections.

3. Follow the Journal"s recommendations in compiling the manuscript and references before publishing.

- We have implemented Journal's recommendations in our review.

We look forward to your response.
Yours faithfully,
Kinga Krawiec 
MD Department of Hematology 
Medical University of Lodz 
2 Ciolkowskiego St 
93-510 Lodz, Poland 
e-mail: [email protected]

Reviewer 2 Report

Krawiec and colleagues present here a comprehensive review on current and future AML treatments aiming at restoring mechanisms of apoptosis. The review is concise and well-structured, and I have only minor comments.

In the simple summary, the verb “to create” (line 5) must be changed for “to establish”, for example.

In figure 1, in the extrinsic pathway, in the blue box representing the death-inducing signaling complex (DISC): are you sure that Caspace 9 can be activated at the DISC? Also, is the Caspace 8/DISC inhibiting c-FLIP or is it the other way around?

In order to improve Table 1, it could be of interest to separate the treatment in trial to patients by adding an additional column.

Author Response

Lodz 30th Aug 2022

Dear Reviewer,
Please find attached the manuscript entitled “Targeting apoptosis in AML: where do we stand?” which we are submitting for consideration for publication as the review article. 
We have introduced your suggestions, and have altered the manuscript accordingly. We feel that the revised manuscript is now much improved and hope that it will now be accepted for publication.

Here are the relevant changes:

  1. In the simple summary, the verb “to create” (line 5) must be changed for “to establish”, for example.

 We have changed this word.

  1. In figure 1, in the extrinsic pathway, in the blue box representing the death-inducing signaling complex (DISC): are you sure that Caspace 9 can be activated at the DISC? Also, is the Caspace 8/DISC inhibiting c-FLIP or is it the other way around?

 You are right. Caspase 9 can’t be activated at the DISC, and Caspace 8/DISC inhibiting c-FLIP is the other way around. We have made the appropriate corrections.

  1. In order to improve Table 1, it could be of interest to separate the treatment in trial to patients by adding an additional column.

We added an additional column in the table.

We look forward to your response.
Yours faithfully,
Kinga Krawiec 
MD Department of Hematology 
Medical University of Lodz 
2 Ciolkowskiego St 
93-510 Lodz, Poland 
e-mail: [email protected]